# Dynamics and variability in the pleiotropic effects of adaptation in laboratory budding yeast populations

Christopher W Bakerlee[1,2†], Angela M Phillips[2†], Alex N Nguyen Ba[2,3], Michael M Desai[2,4,5,6]*

[1]Department of Molecular and Cellular Biology, Harvard University, Cambridge, United States; [2]Department of Organismic and Evolutionary Biology, Harvard University, Cambridge, United States; [3]Department of Cell and Systems Biology, University of Toronto, Toronto, Canada; [4]Department of Physics, Harvard University, Cambridge, Cambridge, United States; [5]NSF-Simons Center for Mathematical and Statistical Analysis of Biology, Harvard University, Cambridge, United States; [6]Quantitative Biology Initiative, Harvard University, Cambridge, Cambridge, United States

*For correspondence:
mdesai@oeb.harvard.edu

†These authors contributed equally to this work

Competing interest: The authors declare that no competing interests exist.

**Abstract** Evolutionary adaptation to a constant environment is driven by the accumulation of mutations which can have a range of unrealized pleiotropic effects in other environments. These pleiotropic consequences of adaptation can influence the emergence of specialists or generalists, and are critical for evolution in temporally or spatially fluctuating environments. While many experiments have examined the pleiotropic effects of adaptation at a snapshot in time, very few have observed the dynamics by which these effects emerge and evolve. Here, we propagated hundreds of diploid and haploid laboratory budding yeast populations in each of three environments, and then assayed their fitness in multiple environments over 1000 generations of evolution. We find that replicate populations evolved in the same condition share common patterns of pleiotropic effects across other environments, which emerge within the first several hundred generations of evolution. However, we also find dynamic and environment-specific variability within these trends: variability in pleiotropic effects tends to increase over time, with the extent of variability depending on the evolution environment. These results suggest shifting and overlapping contributions of chance and contingency to the pleiotropic effects of adaptation, which could influence evolutionary trajectories in complex environments that fluctuate across space and time.

## Introduction

As a population adapts to a given environment, it accumulates mutations that are beneficial in that environment, along with neutral and mildly deleterious 'hitchhiker' mutations. Because these mutations can also affect fitness in other environments, adaptation will tend to lead to pleiotropic fitness changes in other conditions. These pleiotropic consequences of adaptation need not be negative: evolution in one condition can lead to correlated fitness increases in similar environments as well as fitness declines in more dissimilar conditions. It is also natural to expect these consequences to vary over shorter or longer evolutionary timescales. For example, after a sufficiently long time adapting to a single condition, we might expect a population to increasingly specialize to that condition at the expense of its fitness elsewhere.

Numerous laboratory evolution experiments (*Jerison et al., 2020*; *Ostrowski et al., 2005*; *Leiby and Marx, 2014*; *Kinsler et al., 2020*; *Jasmin et al., 2012*; *Novak et al., 2006*; *Meyer et al., 2010*;

*Cooper and Lenski, 2000*; *Bailey and Kassen, 2012*; *Schick et al., 2015*; *Anderson et al., 2011*; *Li et al., 2019*; *Dillon et al., 2016*) as well as empirical studies of natural variation in diverse model systems (*Geiler-Samerotte et al., 2020*; *Wang et al., 2015*; *Hall et al., 2006*; *Mackay and Huang, 2018*) have analyzed the pleiotropic consequences of adaptation. These studies have found examples of specialization, as well as cases of correlated adaptation and the evolution of more generalist phenotypes (*Meyer et al., 2016*; *Hall et al., 2011*; *Duffy et al., 2006*; *Duffy et al., 2007*; *Jerison et al., 2020*; *Li et al., 2019*; *Leiby and Marx, 2014*). Pleiotropic fitness tradeoffs, such as those underlying specialization, can arise from either antagonistic pleiotropy (i.e., direct tradeoffs between the fitness effects of individual mutations across conditions), mutation accumulation (i.e., accumulation of mutations that are neutral in the evolution environment but impose fitness costs in other conditions), or some combination of these phenomena. More complex patterns of correlated fitness changes across conditions, such as those that underlie more generalist phenotypes, can result from more general relationships between fitness effects in different environments. Recent experimental and theoretical work has also analyzed how these distributions of mutational effects across environments can lead to an interplay between chance and contingency in determining both the typical pleiotropic consequences of adaptation and the predictability of these effects (*Jerison et al., 2020*; *Ardell and Kryazhimskiy, 2020*).

The way in which these pleiotropic consequences of adaptation change as populations evolve is less well understood. That is, as a population adapts to a given environment, how steadily and consistently does its fitness change in alternate environments? Do these pleiotropic effects change systematically with time? For example, do fitness tradeoffs tend to become stronger the longer a population adapts to its home environment? And do the pleiotropic consequences of adaptation between replicate lines become more or less similar over time? These questions are critical both for understanding the nature of pleiotropic tradeoffs and for predicting the dynamics and outcomes of evolution in environments that fluctuate across time or space.

Previous studies have shed some light on these questions. For example, *Meyer et al., 2010* reported on changes in phage susceptibility over 45,000 generations of *Escherichia coli* evolution, finding variable yet somewhat consistent trends across six evolved lines. Studying lines from the same evolution experiment, *Leiby and Marx, 2014* found a patchwork of pleiotropic patterns across 12 populations assayed for growth rate in 29 environments at two timepoints. While fitness changed predictably across replicates in some environments, changes were much more variable in others, with mutation rate modifying these patterns. However, these and other studies of the evolutionary dynamics of pleiotropy have been limited to a small number of timepoints, replicate populations, or evolution and assay environments (*Cooper and Lenski, 2000*; *Novak et al., 2006*; *Bailey and Kassen, 2012*). These limitations constrain the degree to which we can make useful inferences about how chance and contingency influence the pleiotropic consequences of adaptation, and how these consequences change over time.

To overcome these limitations, we experimentally evolved hundreds of uniquely barcoded haploid and diploid yeast populations in three environments for 1000 generations. Using sequencing-based bulk fitness assays (BFAs), we assayed the fitness of each evolving population in five environments at 200-generation intervals spanning the 1000 generations of evolution. We then used the resulting data to quantify how the pleiotropic consequences of adaptation unfold in different evolution environments, along with the extent of variation among replicate populations. Our results allow us to investigate differential roles for chance and contingency over evolutionary time, with implications for the outcomes of adaptation in more complex fluctuating environments.

## Results

To study the dynamics of the pleiotropic consequences of adaptation, we experimentally evolved 152 diploid yeast populations for about 1000 generations in one of three different environments (48 populations in rich media (YPD) at 30 °C, 54 populations in YPD +0.2 % acetic acid at 30 °C, and 50 populations in YPD at 37 °C). We chose these environments to facilitate comparisons with previous experimental evolution studies in yeast (e.g., *Nguyen Ba et al., 2019*; *Jerison et al., 2020*), which have used YPD at 30 °C as a rich environment and acetic acid and high temperature to apply distinct types of stress (*Figure 1—figure supplement 2*; *Taymaz-Nikerel et al., 2016*; *Giannattasio et al., 2013*). In addition, we evolved 20 haploid (MATα) yeast populations in YPD at 37 °C; these are a

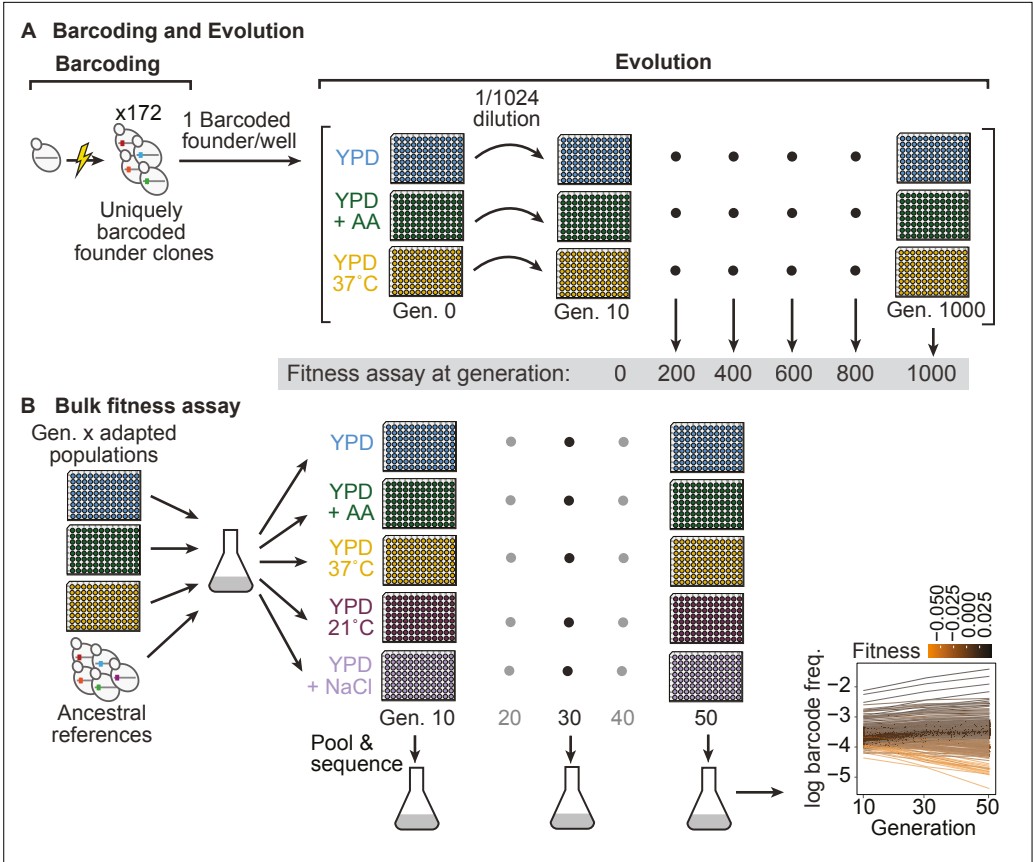

**Figure 1.** Evolution experiment and bulk fitness assay. (**A**) Yeast cells were uniquely barcoded to generate founder clones. Uniquely barcoded founder clones were used to seed individual populations in 96-well plates. Populations were evolved for 1000 generations in three distinct environments: rich media (YPD), rich media at elevated temperature (YPD, 37 °C), and rich media with 0.2 % acetic acid (YPD+ AA), and frozen at 50-generation intervals. Fitness assays were performed at 200-generation intervals. (**B**) Bulk fitness assay of barcoded adapted populations by competitive growth in each evolution environment and two additional environments (YPD, 21 °C and YPD +0.4 M NaCl). Relative fitness of each population was evaluated from the log frequency of the respective barcode sequence over time compared to that of ancestral references, based on assay generations 10, 30, and 50.

The online version of this article includes the following source data and figure supplement(s) for figure 1:

**Source data 1.** Growth curve OD600 and endpoint spot titer measurements; bottleneck sizes for each assay environment.

**Figure supplement 1.** Comparison of technical replicate fitness measurements.

**Figure supplement 2.** Growth curves for ancestors in each assay environment.

---

subset of populations that did not autodiploidize from a larger haploid evolution experiment (see Methods for details).

Each haploid population was founded by a single clone of a putatively isogenic laboratory strain, labeled with a unique DNA barcode at a neutral locus prior to the evolution experiment (*Figure 1A*). Diploid populations were founded by mating uniquely barcoded haploids and selecting for diploids. We then propagated each population for 1000 generations in batch culture, with a 1:2^10 dilution every 24 hr; this corresponds to a population bottleneck size of $10^4$ (*Figure 1A* and *Figure 1—source data 1*; see Methods for details). We froze an aliquot from each population at 50-generation intervals at −80 °C in 8 % glycerol for long-term storage.

After completing the evolution, we revived populations from generations 0, 200, 400, 600, 800, and 1000. We then conducted parallel BFAs (two technical replicates [*Figure 1—figure supplement 1*]) to measure the fitness of each population at each timepoint across five environments (the three evolution environments, YPD +0.4 M NaCl at 30 °C [transfers every 24 hr], and YPD at 21 °C [transfers

every 48 hr]) which exposed the populations to unique osmotic and temperature stresses (*Figure 1—figure supplement 2*). In each BFA, we pooled all populations of a given ploidy from a given generation along with a small number of common reference clones and propagated them for 50 generations (*Figure 1B*). We then sequenced the barcode locus at generations 10, 30, and 50, and we inferred the fitness of each population from the change in log frequency of each corresponding barcode. By exploiting the fact that each population is uniquely barcoded, these BFAs allowed us to estimate the fitness of all 172 populations at each of the five 200-generation intervals in each of the five environments with minimal cost and effort (*Figure 2—source data 1* see Methods for details).

Based on the measured fitness of the generation 0f ancestral populations, we found that some diploid populations had substantially higher ancestral fitness in certain assay environments, likely because they acquired mutations prior to the start of the evolution. To clarify our downstream analyses, we excluded 19 outlier diploid populations whose ancestors differed from the mean ancestral fitness by at least 4 % in at least one environment, leaving us with 133 diploid populations (43 YPD at 30 °C, 48 YPD+ acetic acid, and 42 YPD at 37 °C) and 20 haploid populations (153 populations total). However, we note that the results of all our analyses are very similar when we consider the entire dataset with outliers included (see Figure Supplements).

## Adaptation to the home environment leads to consistent fitness gains and pleiotropic effects

While there is modest variability between replicate populations, adaptation in each environment leads to a consistent increase in fitness in that 'home' environment (*Figure 2*, subplots with bold black borders). As observed in earlier experiments (*Couce and Tenaillon, 2015*), this fitness increase is largely predictable and follows a characteristic pattern of declining adaptability: early rapid fitness gains that slow down over time (p < 0.0001; *Figure 2—figure supplement 6*). There are some differences among evolution environments with respect to this pattern: declining adaptability appears to be especially pronounced in the acetic acid environment, while haploid populations evolved at 37 °C appear not to exhibit this trend, possibly because the fitness gains in this environment were generally minimal.

Adaptation in each evolution environment also led to fitness changes in most other environments (*Figure 2*). In general, these fitness changes tend to have a consistent direction over time for each environment pair. For example, populations adapted to YPD+ acetic acid and YPD at 37 °C steadily gained fitness in the YPD at 30 °C and YPD +0.4 M NaCl environments over time, with the average fitness across populations largely following the same trend seen at home: initial rapid fitness gains followed by slower increases over time. In other instances, fitness gains at home correspond to fitness declines in away environments. For example, populations evolved in YPD+ acetic acid tend to lose fitness in YPD at 21 °C. However, pleiotropic effects are less predictable than the fitness gains in the home environment: we see more variability among replicate lines in away environments, both in the shapes of their fitness trajectories and in their ultimate evolutionary outcomes (see analysis below).

To review the extent of specialization across evolution environments, we summarize the changes in fitness for populations evolved in each environment (*Figure 2B*, *Figure 2—figure supplements 2–5*). Overall, we find that specialization is quite rare, as the majority of populations improve in fitness in each assay environment. The major exception is that a substantial fraction of populations decline in fitness at 21 °C after evolution in other conditions. Additionally, some populations evolved at 37 °C (diploid and haploid) decline in fitness in YPD+ acetic acid, and some populations evolved in YPD+ acetic acid decline in fitness at 37 °C. Importantly, even in the 21 °C environment, specialization is not inevitable, as there are indeed populations evolved in other environments that gain fitness at 21 °C.

To visualize how these pleiotropic effects change over time, we plot these fitness trajectories across pairs of environments (*Figure 3*). This representation of the data shows clear but sometimes subtle differences in patterns of pleiotropy depending on evolution environment and ploidy. For instance, while almost all populations gained fitness in both YPD at 30 °C and YPD+ NaCl, the dynamics of fitness change differed based on evolution environment: populations evolved at 37 °C (orange lines in *Figure 3*) initially made substantial fitness gains in YPD+ NaCl sometimes followed by more significant gains in YPD at 30 °C, whereas the populations evolved in YPD at 30 °C (cyan lines) and YPD+ acetic acid (green lines) only gained substantial fitness in YPD+ NaCl after initial fitness increases in YPD at 30 °C (Figure 3—animation 1).

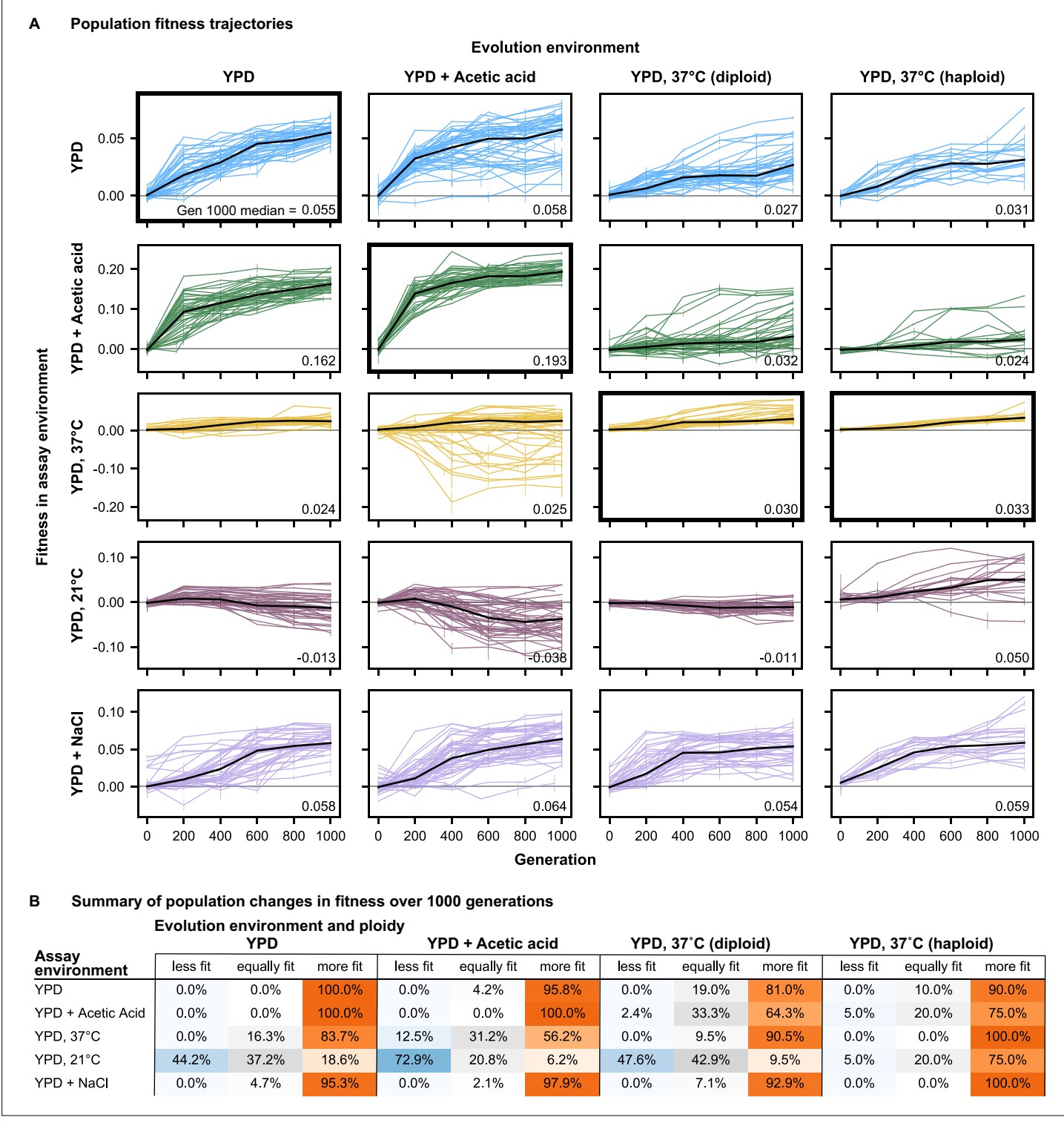

**Figure 2.** Fitness changes over 1000 generations of evolution. (**A**) Population fitness trajectories. Replicate populations for each evolution condition are shown in each column. Environments in which the fitnesses of these populations were assayed are shown in the rows. Plots for which evolution and assay environment are the same are indicated by a bold outer border. The black line in each plot indicates the median fitness. Error bars indicate standard error of the mean. (**B**) Summary of population changes in fitness: generations 0–1000. Populations are categorized according to whether their fitness at generation 1000 is equal to, less than, or greater than their fitness at generation zero. Significance of fitness differences evaluated using one-sided Welch's unequal variances *t*-tests, the number of observations for both fitness values is 2.

The online version of this article includes the following figure supplement(s) for figure 2:

*Figure 2 continued on next page*

*Figure 2 continued*

**Source data 1.** Bulk fitness assay read counts and measured fitnesses.

**Source data 2.** Statistical significance of fitness changes over time.

**Figure supplement 1.** Fitness changes over 1000 generations of evolution for unfiltered data (outliers included).

**Figure supplement 2.** Summary of population changes in fitness: generations 0–200.

**Figure supplement 3.** Summary of population changes in fitness: generations 0–400.

**Figure supplement 4.** Summary of population changes in fitness: generations 0–600.

**Figure supplement 5.** Summary of population changes in fitness: generations 0–800.

**Figure supplement 6.** Changes in fitness early and late in evolution.

Separately, these plots and those in *Figure 2* highlight similarities and differences between the fitness trajectories of populations of different ploidy. Haploids and diploids evolved at 37 °C tend to show quite similar patterns of fitness evolution across alternate environments (*Figure 2*). There are, however, salient differences. For example, comparing fitness in YPD+ NaCl with fitness in YPD at 21 °C reveals haploid trajectories that are both more positive than diploid trajectories in 21 °C and more variable overall (*Figure 3*; Figure 3—animation 1). The divergence of these pleiotropic trajectories is thus contingent on both the evolution environment and an organism's genomic architecture (*Marad et al., 2018*) and associated physiological differences.

## Characteristic environment- and ploidy-specific pleiotropic profiles emerge over time

To understand the diversity of fitness trajectories across environments, we treated the fitness of each population across all five assay environments as a single 'pleiotropic profile'. We then conducted principal component analysis across all these pleiotropic profiles to characterize variation between replicate populations, across different evolution environments, and over time.

In *Figure 4A*, we plot the first two principal components of each pleiotropic profile (which together consistently explain well over half the variance in the data [*Figure 4—figure supplement 2*]) for populations from each of the six measured timepoints (*Figure 4—source data 1*). We see that the populations separate over time into somewhat distinct clusters based on their evolution environment and ploidy. These clusters suggest that evolution in each environment leads to the formation of a characteristic environment- and ploidy-specific pleiotropic profile.

Characteristic pleiotropic profiles can also be observed when running principal component analysis on the complete concatenated (but unordered) fitness data (i.e., with the pleiotropic profile of each population now defined as its fitness across all five assay environments at all six 200-generation timepoints, a total of 30 measurements [*Figure 4—figure supplement 5*]) and plotting data according to the first two components, which explain 30% and 22% of total variance, respectively (*Figure 4B*, *Figure 4—source data 2*). To provide an intuition for the meaning of distance and location in this principal component space, we show home and away environment fitness trajectories for select populations indicated in *Figure 4B, C*. The extent of evolution condition-specific clustering in this two-dimensional PCA is indicative of characteristic pleiotropic profiles (*Figure 4C*), and it appears comparable to that observed in analyses conducted independently for generations 600, 800, and 1000. This is unsurprising given the outsized weighting of later generations in each principal component (*Figure 4—figure supplement 3*).

To more formally quantify the emergence of characteristic pleiotropic profiles over time in *Figure 4A and B*, we developed a simple clustering metric, which counts how many of a given population's five nearest neighbors belong to the same evolution condition on average. We see that the degree of clustering in this two-dimensional space rises appreciably until the 600-generation mark, at which point it plateaus (*Figure 4D*). The observed clustering from generation 200 onward is much greater than expected by chance, as is clustering for the total-data PCA shown in *Figure 4B* (compared to a null expectation constructed by randomly permuting the evolution condition assigned to each population; $p < 0.001$). Note that this trend is consistent when the number of neighbors in the analysis is lowered to 3 or elevated to 10 (*Figure 4—figure supplement 4*). Thus, we observe the rapid emergence and later stabilization of general pleiotropic profiles characteristic to each evolution condition.

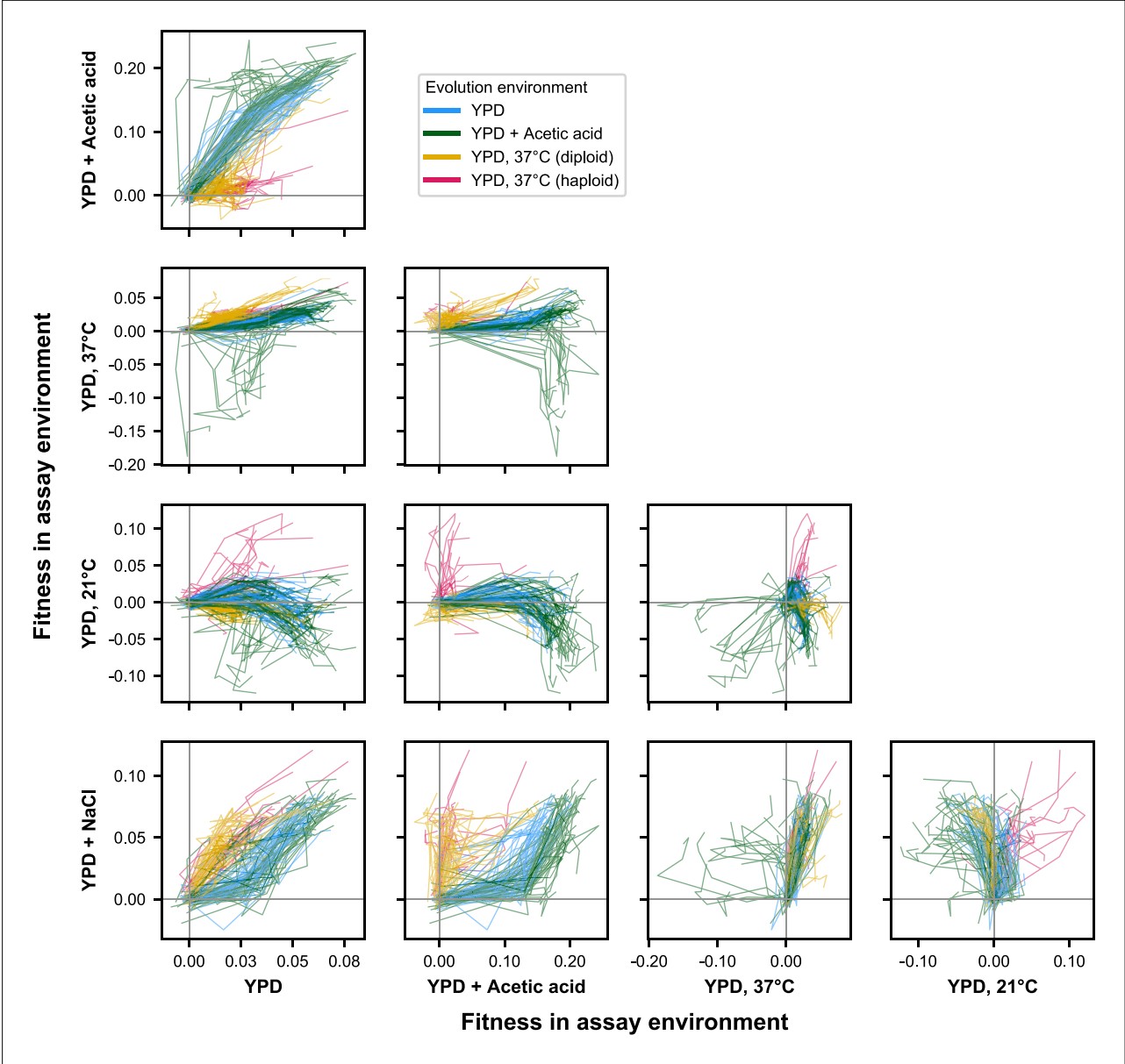

**Figure 3.** E × E evolutionary trajectories over 1000 generations of evolution in a constant environment. Axes correspond to fitness in the indicated assay environments. Colors correspond to evolution condition. Gray vertical and horizontal lines indicate zero fitness relative to an ancestral reference in each environment.

The online version of this article includes the following video and figure supplement(s) for figure 3:

**Figure supplement 1.** E × E evolutionary trajectories over 1000 generations of evolution in a constant environment for unfiltered data (outliers included).

**Figure 3—animation 1.** Animation of Figure 3.

## General trends contain significant variation, which varies with ploidy, environment, and time

Our principal component analysis shows that replicate populations in each evolution condition tend to follow similar trends in fitness changes across environments, leading to characteristic environment-specific pleiotropic profiles. However, it is apparent from *Figures 2 and 3* that there remains significant stochastic variability in the pleiotropic effects of adaptation among populations evolved in the same environment. For instance, populations evolved in the acetic acid environment splay out into all four quadrants when plotting fitness at 37 °C against fitness at 21 °C (*Figure 3*; Figure 3—animation

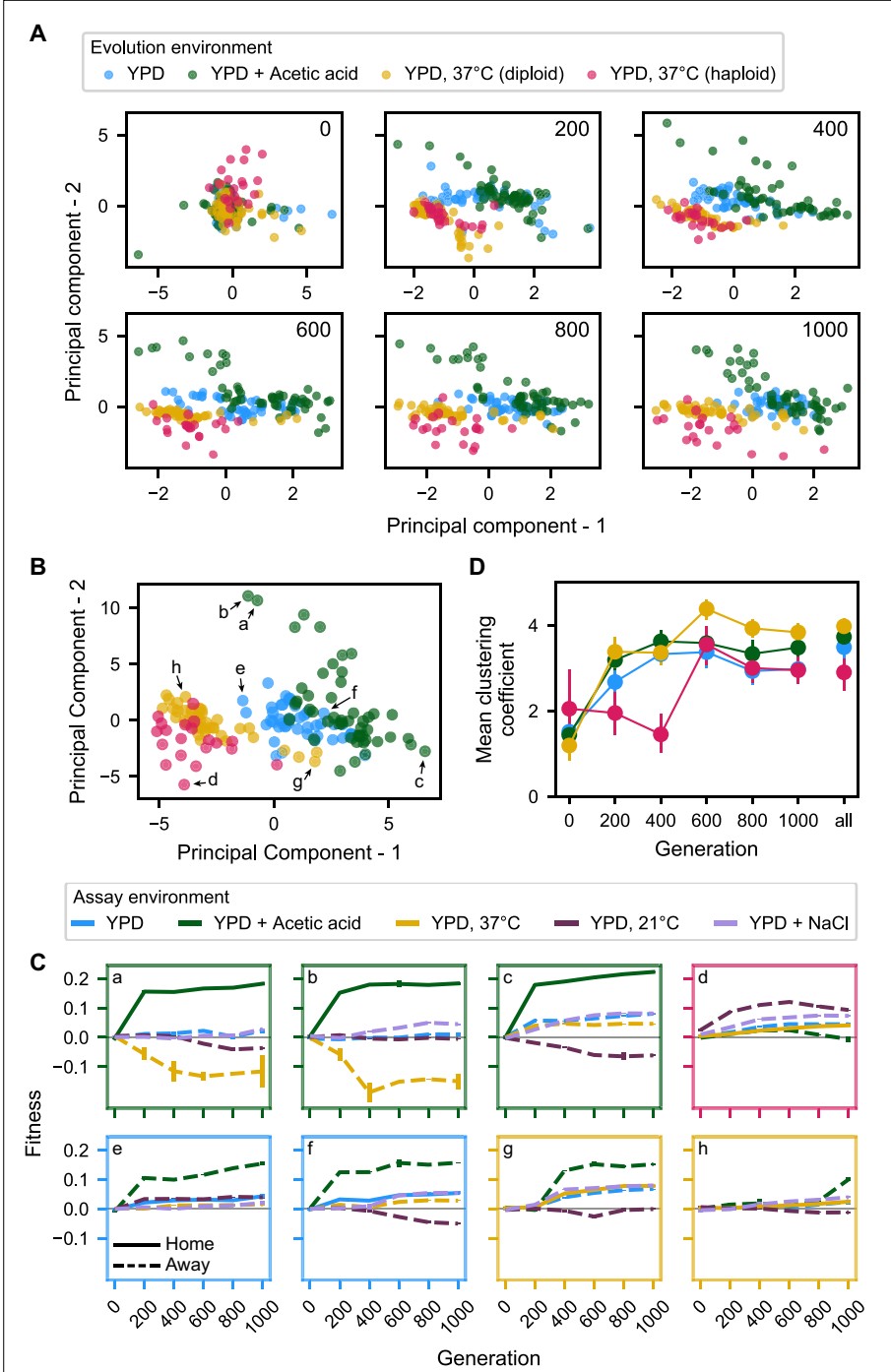

**Figure 4.** Principal component analysis of pleiotropy. (**A**) Principal component analysis of evolving populations, performed independently each 200 generations. The first two PCs are plotted. Populations are colored according to evolution condition. (**B**) Principal component analysis of all populations using all fitness data from across the 1000 generations. The first two PCs are plotted and explain 30% and 22% of the variance, respectively. (**C**) Plots of fitness trajectories in all five assay environments for eight example populations (a–h, identified as points in (**B**)). (**D**) Population clustering in PCA by evolution condition over time. Clustering of each population was quantified as the number of five nearest neighbors that share the same evolution condition, for each 200-generation interval, and across all intervals. Clustering metrics were averaged for each evolution condition to calculate point estimates; error bars represent 95 % confidence intervals of the mean clustering metric, estimated by performing PCA on bootstrapped replicate fitness measurements.

*Figure 4 continued on next page*

*Figure 4 continued*

The online version of this article includes the following source data and figure supplement(s) for figure 4:

**Source data 1.** Principal component analyses presented in *Figure 4A*.

**Source data 2.** Principal component analysis presented in *Figure 4B*.

**Figure supplement 1.** Principal component analysis of pleiotropy for unfiltered data (outliers included).

**Figure supplement 1—source data 1.** Principal component analyses presented in *Figure 4—figure supplement 1A*.

**Figure supplement 1—source data 2.** Principal component analysis presented in *Figure 4—figure supplement 1B*.

**Figure supplement 2.** Variation explained by principal components.

**Figure supplement 3.** Contributions of generation intervals to principal components.

**Figure supplement 4.** Population clustering in PCA as in *Figure 4D* quantified for (**A**) 10 and (**B**) 3 nearest neighbors.

**Figure supplement 5.** Contributions of assay environments to principal components.

---

1). This variability can also be seen in the wide dispersion of populations within clusters in *Figure 4B*, particularly among diploids evolved in the acetic acid environment and at 37 °C.

We find that these patterns of variability are structured, with specific evolution conditions fostering more variable outcomes in certain assay environments (*Figure 5*). For example, populations evolved in YPD+ acetic acid exhibit generally wider variation in home and away environments than populations evolved in other environments. While it is tempting to link this pattern to the large fitness gains these populations make in their home environment, we note that populations evolved in YPD at 30 °C also make significant correlated gains in YPD+ acetic acid without generating such variable results across other assay environments. This suggests that, with respect to the distribution of pleiotropic effects of fixed driver or hitchhiking mutations, paths to higher fitness in YPD+ acetic acid are qualitatively different for the populations evolved in YPD at 30 °C. In another example, while diploid and haploid populations evolved at 37 °C show similar variability in 37 °C, 30 °C, and YPD+ NaCl across the experiment, they experience more variable outcomes in YPD+ acetic acid and 21 °C, respectively. Together, these results suggest that the role for chance in the pleiotropic trajectories of evolving populations is contingent on the condition to which the population is adapted.

In addition, the variation in outcomes is a function of evolutionary time. While variation in fitness at home tends to remain relatively low over the course of 1000 generations (*Figure 5A*, bold black boxes; *Figure 5B*, thick solid lines), variation in away environments generally (if haltingly) increases over time, with a few exceptions. In other words, selection appears to suppress variation among trajectories in the home environment, at least on the timescales studied. To assess the statistical significance of these differences in variance, we used a one-tailed variant of a Brown–Forsythe test to perform pairwise comparisons of home and away fitness variance among replicate lines evolved in a given condition at each evolution timepoint. Of the 80 nonancestral pairwise comparisons, over half (48/80) indicated significantly greater variance in the away environment (at a threshold of $p < 0.05$) and only six showed significantly greater variance at home (*Figure 5—figure supplement 2*).

The role of stochasticity and temporal shifts in pleiotropic dynamics also can be seen in the relative nonmonotonicity of fitness trajectories in away environments compared to home environments. To assess nonmonotonicity, we interpolated fitness at 500 generations for each population in each assay environment and compared the 0- to 500-generation and 500- to 1000-generation fitness changes. Trajectories were considered nonmonotonic if fitness changes in these intervals were in opposite directions (*Figure 6A*, see shaded quadrants), reflecting pleiotropic effects that change in sign over time. We find that populations rarely possess clearly nonmonotonic trajectories in their home environment, whereas they much more commonly possess clearly nonmonotonic trajectories in away environments (4/153 [2.6%] home and 102/612 [16.7%] away trajectories, respectively; $p < 0.0001$, $\chi^2$ test) (*Figure 6B*). Many but not all of these monotonic trajectories (72/102, or 71%) reflect initially positive pleiotropic effects that become negative in the second half of the experiment, as we might expect if a population increasingly specializes to its home environment over time.

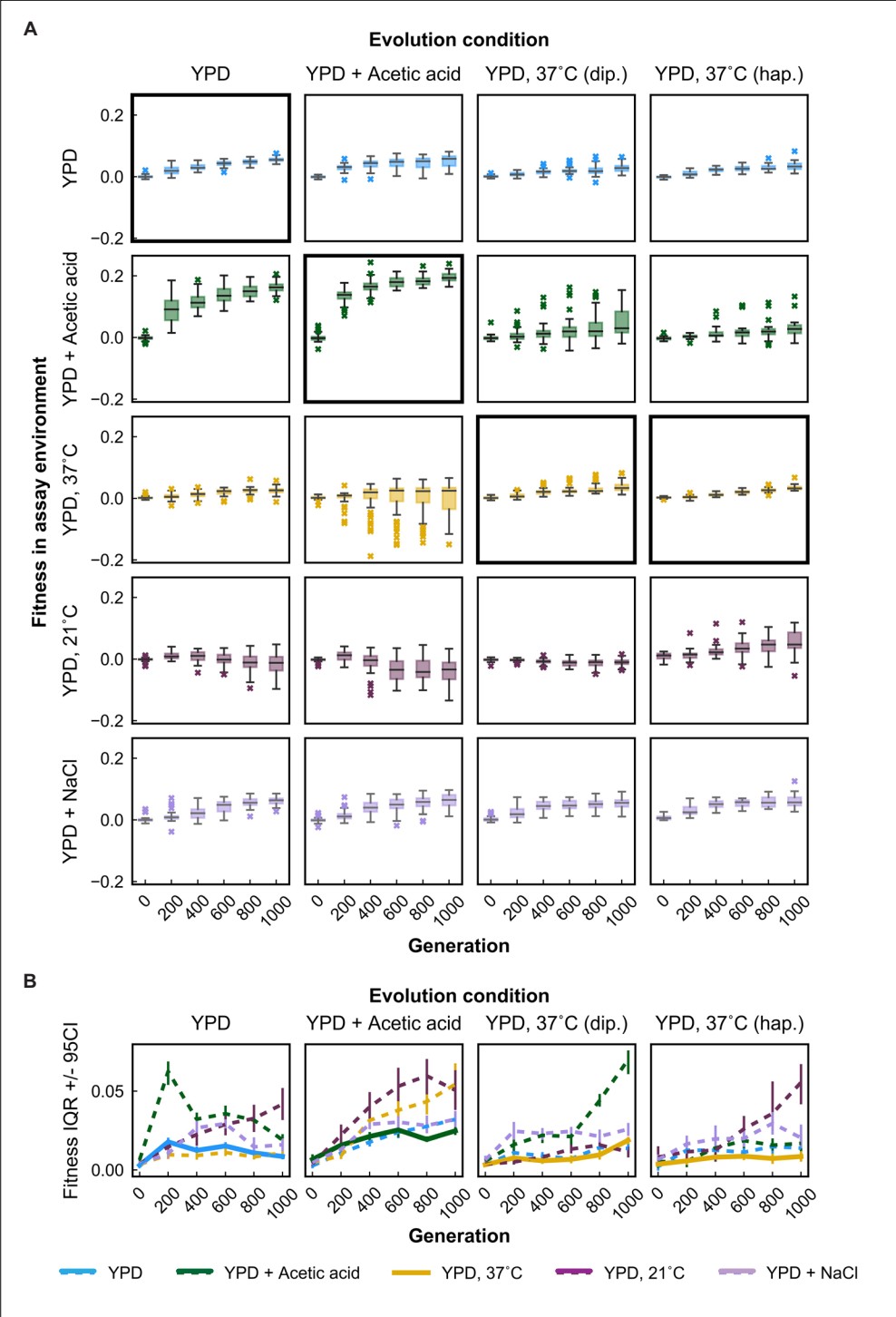

**Figure 5.** Variability in fitness over time. (**A**) Box plots summarizing population mean fitness over time for each evolution condition (columns) in each assay environment (rows). Line, box, and whiskers represent the median, quartiles, and data within 1.5 × IQR (interquartile range), respectively; outlier populations beyond whiskers are shown as points. (**B**) IQR from box plots in (**A**) are plotted as a function of time for each evolution condition and assay environment. IQR for fitness measured in home and away environments are represented by solid and dashed lines, respectively. Error bars represent 95 % confidence intervals of IQR calculated from bootstrapped replicate fitness measurements.

The online version of this article includes the following figure supplement(s) for figure 5:

*Figure 5 continued on next page*

*Figure 5 continued*

**Source data 1.** Brown–Forsythe test statistics.

**Figure supplement 1.** Variability in fitness over time for unfiltered data (outliers included).

**Figure supplement 2.** Statistical test of difference in variance between home, away environments.

## Discussion

To characterize the dynamics of pleiotropy during adaptation, we evolved hundreds of diploid and haploid yeast populations in three environments for 1000 generations, and assayed their fitness in these and two other environments at 200-generation intervals. Our results offer insight into how pleiotropic effects emerge and change on an evolutionary timescale. Consistent with earlier work, we observe repeatable fitness trajectories across many replicate populations in their home environments, which follow a pattern of initial rapid fitness gains followed by declining adaptability over time. Replicate populations also tend to follow consistent fitness trajectories in away environments, whether gaining or losing fitness on average. Looking across populations and environments, characteristic patterns of pleiotropy specific to each evolution condition emerge rapidly and stabilize within about 600 generations.

Despite these characteristic patterns, we also observe ample variability within these trends. Examining the fitness trajectories of populations individually, we find that about 17 % of away-environment trajectories are nonmonotonic, compared to just 3 % of home-environment trajectories. This nonmonotonicity is indicative of the sequential establishment of mutations with opposing pleiotropic effects

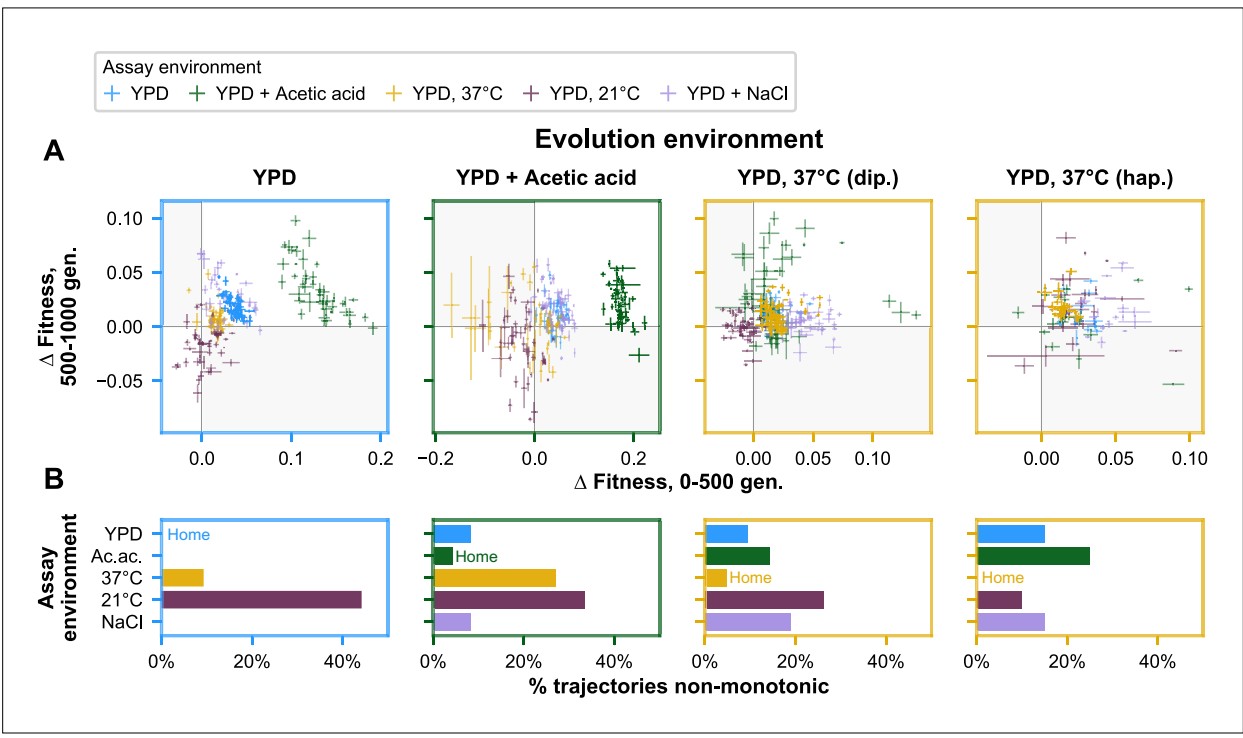

**Figure 6.** Nonmonotonicity in evolutionary trajectories. (**A**) Each panel shows, for each of the five assay environments, the change in fitness over the first 500 (*x*-axis) and second 500 (*y*-axis) generations of evolution of each population in a given evolution environment. Error bars correspond to standard error. Populations that fall in shaded quadrants have trajectories that are nonmonotonic. Points corresponding to fitness in the home environment are colored more opaquely than points corresponding to fitness in away environments, and panel borders have been colored to match the home environment. Fitness at generation 500 has been interpolated. (**B**) Each panel corresponds to a given evolution environment and shows the proportion of populations evolved in that environment that exhibit clearly nonmonotonic fitness trajectories in (**A**). 'Clearly nonmonotonic' trajectories are those populations (points) in (**A**) that fall in the gray quadrants and whose error bars (one standard error in either direction) do *not* span either the *x*- or *y*-axis. As in (**A**), bars corresponding to the home environment are colored more opaquely than bars corresponding to away environments.

The online version of this article includes the following figure supplement(s) for figure 6:

**Figure supplement 1.** Nonmonotonicity in evolutionary trajectories for unfiltered data (outliers included).

in these populations. Meanwhile, across replicate populations, there is substantial variability in the pleiotropic consequences of evolution in each condition. Consistent with past work, we observe more variability in away than in home environments at the end of the experiment (*Travisano and Lenski, 1996*; *Ostrowski et al., 2005*). However, our results also reveal how populations can follow very different trajectories in arriving at these endpoint fitnesses. Diverse away-environment trajectories manifest as changes in the variance among replicate populations over time, with a general tendency for variance to increase over the course of the experiment.

Together, patterns of pleiotropy along with variability among replicate populations suggest an important and dynamic role for chance and contingency in the fates of populations evolving in environments that fluctuate in space and time. Whether populations trend toward specialist or generalist phenotypes will not simply reflect physiological constraints (*Bono et al., 2017*; *Jerison et al., 2020*). Rather, as we observe, mutational opportunities to move toward higher or lower fitness in alternate environments may be accessible at all times. Thus, the emergence of specialism or generalism will be a product of both the distribution of pleiotropic effects of mutations that establish and dynamical factors that influence the timescale, sequence, and likelihood of their fixation (e.g., epistasis, ploidy, clonal interference, mutation rate, and population size). For instance, while previous studies observe substantial specialization in high-salt environments (*Jerison et al., 2020*), here we observe general improvement in fitness in high-salt across evolution environments (*Figure 2*), which may be attributable to differences in the strain background or ploidy, the salt concentration used, or some combination of these factors.

Furthermore, the timescale over which pleiotropic effects emerge and change will interact with patterns of environmental fluctuations to determine evolutionary outcomes. In the conditions studied here, we observe that pleiotropic profiles generally emerge early and stabilize by 600 generations. Independent of other dynamical consequences of the rate of environmental change (*Cvijović et al., 2015*), it is therefore likely that fluctuations on longer timescales (e.g., longer than 600 generations in this system) will lead to qualitatively different outcomes than fluctuations on shorter timescales. Our data show that both the average and variance in these outcomes will also depend critically on the specific sequence of environments experienced by a population.

These results underscore the need for further empirical and theoretical work to understand patterns of pleiotropic effects over time and their effects on evolutionary trajectories. Additional experiments will be required to describe how general pleiotropic trends and variability within these trends arise and shift across a wider array of environments, as well as in different model systems. Likewise, studies of pleiotropy in populations evolved for longer periods, such as those described by *Johnson et al., 2021*, may provide a richer perspective on the repeatability, diversity, and stability of pleiotropic trajectories. Finally, this work motivates further theoretical inquiry into how the dynamics and variability of pleiotropic effects will interact with other important parameters – such as patterns of environmental fluctuation, mutation rate, sexual recombination, and the underlying distributions of fitness effects – to influence evolutionary outcomes. Integrating empirical datasets like the one presented here with such theoretical insight will enable better prediction of adaptation in complex environments.

## Materials and methods

**Key resources table**

| Reagent type (species) or resource | Designation | Source or reference | Identifiers | Additional information |
|---|---|---|---|---|
| Strain, strain background (*Saccharomyces cerevisiae*) | YCB140B | This paper | *MATa, his3Δ1, leu2Δ0, lys2Δ0, RME1pr::ins-308A, ycr043cΔ0::NatMX, can1::STE2pr_SpHIS5_STE3pr_LEU2, ybr209w::GAL10pr-CRE, trp1Δ, URA3::STE5pr_URA3, HO::CgTRP1* | |
| Strain, strain background (*Saccharomyces cerevisiae*) | YCB137A | This paper | *MATα, his3Δ1, leu2Δ0, lys2Δ0, RME1pr::ins-308A, ycr043cΔ0::NatMX, can1::STE2pr_SpHIS5_STE3pr_LEU2, ybr209w::GAL10pr-CRE, trp1Δ, URA3::STE5pr_URA3, HO::CgTRP1* | |
| Recombinant DNA reagent | Barcoding plasmid landing pad 1 | This paper | | Plasmid map in *Supplementary file 2* |

*Continued on next page*

*Continued*

| Reagent type (species) or resource | Designation | Source or reference | Identifiers | Additional information |
|---|---|---|---|---|
| Recombinant DNA reagent | Barcoding plasmid landing pad 2 | This paper | | Plasmid map in *Supplementary file 3* |
| Sequence-based reagent | Illumina sequencing primers | IDT | | Sequences listed in *Supplementary file 4* |
| Peptide, recombinant protein | Zymolyase 20T | Nacalai Tesque | Zymolyase 20T | |
| Software, algorithm | Custom code | This paper | | https://github.com/amphilli/pleiotropy-dynamics |

## Strain generation

Strains in this study are derived from YAN404 and YAN407 (*Nguyen Ba et al., 2019*), which were constructed on the BY4742 background (S288C: *MATα*, *his3Δ1*, *ura3Δ0*, *leu2Δ0*, *lys2Δ0*) to add the *RME1*pr::ins-308A mutation, meant to improve transformation efficiency in both the *MAT**a*** and *MATα* cell types. Several additional modifications were made to enable proper barcoding, mating, and selection, as stated in *Supplementary file 1*. Ultimately, YCB140B and YCB137A (and YCB140B × YCB137A mated diploids) were used to found the populations evolved in this experiment.

## Barcode plasmid design and integration

Our barcoding system uses two different landing pad types, hereafter referred to as type 1 and type 2. Both plasmids had a pUC origin and ampicillin resistance cassette in the vector backbone. The inserts into this 1998 bp backbone were 6728 and 6384 bp, respectively, with ~450 bp homology to the regions flanking the *CgTrp1* in the *HO* locus on either side. Between these flanking regions were modified versions of the *KanMX* and *CAN1* genes, as well as a *ccdB* gene that is toxic to sensitive *E. coli* strains. Many other components, including lox sites, artificial introns, and unexpressed *TRP1* genes, were also present in these plasmids, and the entirety of the annotated plasmids can be viewed in *Supplementary files 2 and 3*. These extraneous elements – both in the plasmids and in our strain backgrounds – were included to enable capabilities that ultimately were not harnessed for the purposes of this study, such as mating, sporulation, and the inducible and selectable Cre-driven recombination of barcodes.

To generate diversely barcoded plasmid libraries, we cloned oligonucleotides containing random nucleotides into the type 1 and type 2 plasmids via a Golden Gate reaction (*Engler et al., 2008*). This reaction replaced the *ccdB* gene in the plasmid. The barcoded plasmids were transformed via electroporation into *ccdB*-sensitive *E. coli*. Barcoded plasmids were then purified from these transformants using the Geneaid Presto Mini Plasmid Kit (Cat. No. PDH300).

To barcode ancestral YCB137A and YCB140B strains, we took advantage of PmeI restriction endonuclease sites on either side of the *HO* homology regions of the plasmid, cutting and transforming (*Gietz, 2015*) into the *HO* locus and replacing the *CgTRP1* gene.

To select for successful haploid yeast transformants, we used 200 µg/ml G418 (GoldBio, G-418), following up with a screen in SD-Trp (1.71 g/l Yeast Nitrogen Base Without Amino Acids and Ammonium Sulphate [Sigma-Aldrich, Y1251], 5 g/l ammonium sulfate [Sigma-Aldrich, A4418], 20 g/ʟ dextrose [VWR #90000-904], 0.1 g/ʟ L-glutamic acid [Sigma-Aldrich, G1251], 0.05 g/ʟ L-phenylalanine [Sigma-Aldrich, P2126], 0.375 g/ʟ L-serine [Sigma-Aldrich, S4500], 0.2 g/l L-threonine [Sigma-Aldrich, T8625], 0.01 g/l myo-Inositol [Sigma-Aldrich, I5125], 0.08 g/l adenine hemisulfate salt [Sigma-Aldrich, A9126], 0.035 g/l L-histidine [Sigma-Aldrich, H6034], 0.11 g/l L-leucine [Sigma-Aldrich, L8000], 0.12 g/l L-lysine monohydrate [Acros Organics, CAS: 39665-12-8], 0.04 g/l L-methionine [Sigma-Aldrich, M9625], 0.04 g/l uracil [Sigma-Aldrich, U1128]). After ~25 generations of selection in liquid media, strains auxotrophic for tryptophan and resistant to G418 were arrayed into plates for experimental evolution.

Other successful transformants (of the same landing pad type) were mated to form diploids, which were selected for resistance to 300 µg/ml hygromycin B (GoldBio, H-270), 100 µg/ml nourseothricin sulfate (GoldBio, N-500), 200 µg/ml G418, and 1 mg/ml 5-fluoroorotic acid monohydrate (Matrix

Scientific, CAS: 220141-70-8) in S/MSG D media (1.71 g/l Yeast Nitrogen Base Without Amino Acids and Ammonium Sulphate, L-glutamic acid monosodium salt hydrate [Sigma-Aldrich, G1626], 20 g/l dextrose, 0.1 g/l L-glutamic acid, 0.05 g/l L-phenylalanine, 0.375 g/l L-serine, 0.2 g/l L-threonine, 0.01 g/l myo-Inositol, 0.08 g/l adenine hemisulfate salt, 0.035 g/l L-histidine, 0.11 g/l L-leucine, 0.12 g/l L-lysine monohydrate, 0.04 g/l L-methionine, 0.04 g/l uracil, 0.08 g/l L-tryptophan [Sigma-Aldrich, T0254]) for ~25 generations prior to arraying into 96-well plates alongside haploids for experimental evolution.

## Experimental evolution

Barcoded yeast were used to found 192 *MAT*a, 192 *MAT*α, and 162 diploid populations for evolution, respectively (though most haploid populations were excluded from further analysis due to the fixation of autodiploids). Each population was founded by a uniquely barcoded single colony or uniquely barcoded colonies that were then mated to form a diploid (see 'Strain generation'), and was subsequently propagated in a well of an unshaken flat-bottom polypropylene 96-well plate in one of three conditions: YPD (1 % Bacto yeast extract [VWR #90000-726], 2 % Bacto peptone [VWR #90000-368], 2 % dextrose) at 30 °C, YPD at 37 °C, and YPD +0.2 % acetic acid (Sigma Aldrich #A6283) at 30 °C (128 µl/well). Each 96-well plate contained diploid and haploid populations of both mating types (with each mating type occupying one side of the plate) and five empty wells to monitor for potential cross contamination. With the exception of the YPD at 37 °C condition, the evolution conditions were arranged in a checkered pattern on each 96-well plate to minimize potential plate effects. Daily $1:2^{10}$ dilutions (bottleneck ~$10^4$ cells) were performed using a Biomek-FX pipetting robot (Beckman-Coulter) after thorough resuspension by shaking on a Titramax 100 orbital plate shaker at 1200 rpm for at least 1 min. Populations underwent daily transfers for ~1000 generations (~10 generations/day); every 50 generations, populations were mixed with glycerol to a final concentration of 8 % for long-term storage at −80 °C. No contamination of blank wells was observed over the course of the evolution experiment. One of the 96-well plates was dropped at generation 170 and evolution was resumed by thawing and reviving populations from the generation 150 archive; thus, all future archives of populations on this plate lagged 40 generations behind the populations on all other plates.

## Population growth curve and bottleneck size measurements

Growth curves were observed and population bottleneck sizes were determined for two haploid and two diploid ancestral clones in each of the evolution and assay environments. Clones were first preconditioned in each environment for 20 generations (except for the 21 °C environment, in which preconditioning was performed for 10 generations). Following preconditioning, cultures were serially diluted and spotted onto YPD-agar to determine the population bottleneck size (*Figure 1—source data 1*). To measure growth rate, the same preconditioned cultures were diluted $1:2^{10}$ in technical duplicate in 96-well plates and shaken on a Titramax 100 orbital plate shaker at 1200 rpm for 1 min, as in the evolution experiment (see above). Plates were then sealed with a breathable membrane (VWR #60941-086) and loaded into a Biotek Epoch 2, and relative absorbance measurements (600 nm) were made for 24 hr (48 hr for the 21 °C environment) at an interval of 20 min, preceded by 2 min of linear shaking at maximum speed. Following the 24 (or 48)-hr period of absorbance measurements, cultures were diluted and spotted onto YPD-agar to quantify the final cell density. Relative absorbance measurements and cell densities are provided in *Figure 1—source data 1* and the corresponding growth curves are plotted in *Figure 1—figure supplement 2*. We note that for the 21 °C evolution environment, the growth rate was actually measured at 24 °C due to practical constraints.

## Nucleic acid staining for ploidy

Populations frozen at generation 1000 of the evolution experiment were thawed and revived by diluting $1:2^5$ in YPD. The following day, saturated cultures were diluted 1:20 into 120 µl of sterile water in round-bottom polystyrene 96-well plates. Plates were centrifuged at 3000 × *g* for 3 minutes, the supernatant was removed, and cultures were resuspended in 50 µl sterile water. 100 µl of ethanol was added to each well, the cultures were mixed thoroughly and placed at 4 °C overnight. The following day, the cultures were centrifuged, the ethanol solution was removed, and 65 µl RNase A (VWR #97062-172) solution (2 mg/ml RNase A in 10 mM Tris–HCl, pH 8.0 + 15 mM NaCl) was added to each well and the cultures were incubated at 37 °C for 2 h. Then 65 µl of 300 nM SYTOX green (Thermo

Fisher Scientific, S-34860) was added to each well and the cultures were mixed and incubated at room temperature in the dark for 30 min. Fluorescence was measured by flow cytometry on a BD LSRFortessa using the FITC channel (488 nm). Ploidy was assessed by comparing the fluorescence distributions of evolved populations to known haploid and diploid controls of the same strain. By generation 1000, all 192 *MAT*a populations had autodiploidized, and 172 of the *MAT*α populations had autodiploidized, as judged by the absence of a clear haploid peak. Only the remaining 20 haploid *MAT*α populations were included in the BFAs described below.

## Bulk fitness assays

Populations frozen at generations 0, 200, 400, 600, 800, and 1000 of the evolution experiment were thawed by diluting $1:2^5$ in YPD. The following day, once these cultures had grown to saturation, equivalent volumes of each population were pooled by ploidy for each generation (12 pools total). For the haploid populations, evolved populations were only pooled if they were verified to be haploid at the end of the evolution experiment (see 'Nucleic acid staining for ploidy'). Each of the haploid pools was spiked with five uniquely barcoded ancestral reference strains of the same mating type at 4× the volume of each evolved population; each of the diploid pools was spiked with 10 reference strains at 4× the volume of each evolved population. The resulting pools comprised time point zero for the BFA and were diluted $1:2^{10}$ in the appropriate media (described below) and divided between 16 wells (128 μl/well) of flat-bottom polypropylene 96-well plates. The BFA was performed in each of the three evolution environments (YPD at 30 °C, YPD at 37 °C, and YPD +0.2 % acetic acid at 30 °C), in addition to two novel environments (YPD at 21 °C and YPD +0.4 M NaCl at 30 °C). The 16 wells of each pool comprised two technical replicates of 8 wells. Every 24 hr (or every 48 hr in the case of the YPD 21 °C environment) the populations were resuspended by shaking on a Titramax 100 orbital plate shaker at 1,200 rpm for at least 1 min and the contents of the eight wells constituting each replicate were combined, mixed, and diluted $1:2^{10}$ into eight new wells using a Biomek-FX pipetting robot (Beckman-Coulter). This split-pool strategy was designed to mimic the evolution conditions while maintaining sufficient diversity for bulk fitness measurements. At BFA timepoints 0, 10, 30, and 50 generations, 1 ml of the diploid pool was combined with 200 μl of the haploid pool for each generation, this culture was centrifuged at 21,000 × *g* for 1 min, the supernatant was removed, and the pellet was stored at −20 °C for downstream DNA extraction and sequencing.

## Sequencing library preparation

Genomic DNA was extracted from cell pellets using zymolyase-mediated cell lysis (5 mg/ml Zymolyase 20T (Nacalai Tesque), 1 M sorbitol, 100 mM sodium phosphate pH 7.4, 10 mM EDTA, 0.5 % 3-(*N*,*N*-dimethylmyristylammonio)propanesulfonate (Sigma T7763), 200 μg/ml RNase A, 20 mM DTT), binding on silica columns (IBI scientific, IB47207) with 4 volumes of guanidine thiocyanate (4.125 M guanidine thiocyanate, 100 mM MES pH 5, 25 % isopropanol, 10 mM EDTA), washing with wash buffer 1 (10% guanidine thiocyanate, 25 % isopropyl alcohol, 10 mM EDTA) and wash buffer 2 (20 mM Tris–HCl pH 7.5, 80 % ethanol), and eluting in 50 μl 10 mM Tris pH 8.5, as previously described (*Nguyen Ba et al., 2019*). Two rounds of PCR were performed to generate amplicon sequencing libraries for sequencing the barcode locus. In the first round of PCR, the barcode locus was amplified with primers containing unique molecular identifiers (UMI), generation-specific inline indices, and partial Illumina adapters (see *Supplementary file 4* for primer sequences). This 20 μl 10-cycle PCR reaction was performed using Q5 polymerase (NEB M0491L) following the manufacturer's guidelines, using 10 μl (~250 ng) of gDNA as template, annealing at 54 °C, and extending for 45 s. The first-round PCR products were then purified using one equivalent volume of DNA-binding beads (Aline Biosciences PCRCleanDX C-1003-5) and eluting in 33 μl 10 mM Tris pH 8.5. In the second-round PCR, the remainder of the Illumina adapters and sample-specific Illumina indices were appended to the first-round PCR products (see *Supplementary file 4* for primer sequences). The second-round PCR was performed using Kapa HiFi HotStart polymerase (Kapa Bio KK2502) following the manufacturer's guidelines for a 25 μl reaction, using 17.25 μl of first-round PCR product, annealing at 63 °C and extending for 30 s for 26 cycles. The second-round PCR products were then purified using one equivalent volume of DNA-binding beads and eluting in 33 μl 10 mM Tris pH 8.5. Following bead cleanup, the concentration of the PCR products was quantified using the Accugreen High Sensitivity dsDNA Quantitation Kit (Biotium 31068).

Sequencing libraries were then pooled equally and sequenced on a NextSeq500 Mid flow cell (150 bp single-end reads).

## BFA barcode enumeration and fitness inference

Lineage fitnesses were inferred from the concatenated sequencing data yielded by two separate NextSeq500 Mid flow cells (150 bp single-end reads). The second of these two runs allowed for deeper sequencing of specific BFA timepoints to enable superior determination of barcode frequencies associated with less fit lineages in certain environments. The second run also allowed sequencing of libraries that were omitted from the first run.

Once fastq files were concatenated, barcode information was extracted as described below. However, in addition to subjecting the barcode regions to error-tolerant 'fuzzy' matching based on regular expressions, we allowed for fuzzy matching of the epoch-specific inline indices. For the indices, we applied a list of decreasingly strict regular expressions, looking for exact matches, then one mismatch, then two mismatches. For the indices associated with epochs 6, 8, and 10, which were longer than the indices associated with epochs 0, 2, and 4, we allowed up to three mismatches.

Then, as with the barcode association mapping, we used a previously described 'deletion-error-correction' algorithm (*Johnson et al., 2019*) to correct errors in barcode sequences induced by library preparation and sequencing.

To check for cross-contamination between wells during library preparation and index-hopping during sequencing, we searched for reads where the inline index was inconsistent with the associated pairs of Illumina indices. In almost all cases, we found little evidence of cross contamination ( << 1%). In one case, corresponding to landing pad type 2 of the 30 °C replicate 2 BFA 10-generation timepoint for generation-1000 populations, we found that 11,484 of the 258,462 reads (4.4%) included the inline index associated with the generation-200 populations. We removed all apparently cross-contaminating reads from our analysis.

Then, we summed reads associated with all barcodes in a given population, since some populations contained more than one unique barcode (or, in the case of diploids, more than two unique barcodes). In addition, some barcodes were present in the BFAs that could not be confidently assigned to a single well, representing 0.3 % of all reads. These were summed together and retained in the dataset.

To determine the fitness of each population over time and across environments and technical replicates, we measured the log-frequency slope for each population in two intervals: between assay timepoints 10 and 30 and between timepoints 30 and 50 generations. Frequencies were calculated separately for each landing pad type. We scaled these values of fitness ($s$) by subtracting out the corresponding median log-frequency slope of a set of between 2 and 5 reference ancestral populations of each ploidy and landing pad type, which were included in every BFA to allow comparisons of fitness across the evolutionary time course. The source data file indicates these reference populations. For a given BFA and interval, $s$ values only were calculated this way if the mean number of reads for the reference populations was greater than 5. If not, these intervals were excluded from subsequent analysis.

To determine $s$ values for each population in each environment at each generation, interval-specific $s$ estimates were averaged. Then, $s$ estimates from each of the two technical replicates were averaged, producing a final $s$ estimate. The standard error of this final $s$ estimate was calculated from the two technical replicate $s$ estimates.

To clarify our downstream analyses, we excluded 19 outlier diploid populations whose ancestors differed from the mean ancestral fitness by at least 4 % in at least one environment. We believe we see such divergent ancestral fitness values due mutations that emerged during the process of selecting colonies, mating, and performing purifying selection for ~50 generations on barcoded transformants immediately prior to evolution.

To account for the offset in plate two progress through evolution, plate two population fitness estimates for 200, 400, 600, and 800 generations were linearly interpolated from fitnesses on either side, e.g., gen 200 fitness inferred from gen 160 and gen 360 fitnesses. Fitness estimates for gen 1000 were extrapolated linearly from gen 760 and gen 960 fitnesses. The standard error of the $s$ estimate for gen 160 was used for gen 200 fitness, the standard error of $s$ for gen 360 was used for gen 400 fitness, and so on.

## Barcode association

To map barcodes to wells of the evolution experiment, we pooled ancestral strains in equal volumes from across the eight evolution plates, creating three sets of pools: column-specific pools ($n$ = 12), row-specific pools ($n$ = 8), and plate-specific pools ($n$ = 8). We then lysed portions of these pools by diluting in yeast lysis buffer 1 mg/ml Zymolyase 20T, 0.1 M sodium phosphate buffer pH 7.4, 1 M sorbitol, 10 mg/ml SB3-14 (3-($N$,$N$-dimethylmyristylammonio)propanesulfonate [Sigma T7763]) at 37 °C for 1 hr and 95 °C for 10 min. Two rounds of PCR were then performed to generate amplicon sequencing libraries for sequencing the barcode locus (both landing pad versions). In the first round, the barcode locus was amplified via a 10-cycle PCR reaction with Kapa HiFi HotStart polymerase (Kapa Bio KK2502), annealing at 58 °C for 30 s and extending at 72 °C for 30 s, with a final 10 min extension. PCR products were then purified using one equivalent volume of DNA-binding beads and eluting in 20 µl water. Following bead purification, a second-round PCR reaction was performed using 1.5 µl of each of a unique pair of Illumina indices (see *Supplementary file 4* for primer sequences) with Kapa HiFi Hotstart ReadyMix (2× ) in a 15 µl reaction, with 4.5 µl of first-round PCR product as template, annealing at 61 °C and extending for 30 s for 30 cycles. The second-round PCR products were then purified using 0.8× DNA-binding beads (Aline Biosciences PCRClean DX C-1003-5), washed 2× with 80 % ethanol and eluted in 50 µl of molecular biology-grade water. Following bead cleanup, the concentration of the second-round PCR products was quantified using the Accugreen High Sensitivity dsDNA Quantitation Kit (Biotium 31068). These libraries were then normalized, pooled, and sequenced on a NextSeq500 High flow cell (150 bp paired-end reads).

To extract barcode information from sequencing reads, we followed *Johnson et al., 2019*, using a list of decreasingly strict regular expressions (using the python regex module https://pypi.org/project/regex/). For landing pad 1, this was:

> '(TCTGCC)(\D{22})(CGCTGA)',
> '(TCTGCC)(\D{20,24})(CGCTGA)',
> '(TCTGCC){e ≤ 1}(\D{22})(CGCTGA){e ≤ 1}',
> '(TCTGCC){e ≤ 1}(\D{20,24})(CGCTGA){e ≤ 1}'.
> For landing pad 2, this was:
> '(TCTCTG)(\D{22})(AGTAGA)',
> '(TCTCTG)(\D{20,24})(AGTAGA)',
> '(TCTCTG){e ≤ 1}(\D{22})(AGTAGA){e ≤ 1}',
> '(TCTCTG){e ≤ 1}(\D{20,24})(AGTAGA){e ≤ 1}'.

Then, after parsing and tallying barcodes in each sequencing library, we used the 'deletion-error-correction' algorithm described by *Johnson et al., 2019* to correct errors in barcode sequences induced by library preparation and sequencing.

To triangulate the position of each barcode across the eight plates, for each error-corrected barcode that appeared in the sequencing data, we tabulated which barcodes were present in which libraries, and how many reads were associated with each barcode in each library. These data allowed us to determine the wells in which barcodes belonged.

## Changes in fitness analysis

To summarize changes in fitness at each generation since the beginning of the experiment, we assessed the fraction of populations in each assay environment and evolution condition that had increased in fitness, decreased, or remained the same. To categorize a population in one of these three categories, we performed a Welch's unequal variances $t$-test comparing the fitness of that population in a given assay environment at 0 generations to the fitness of that population at 200, 400, 600, 800, or 1000 generations. Since each fitness measurement is the result of two independent technical replicate measurements, we treated each as the mean of two observations. If the fitness at the later timepoint was greater than the ancestral fitness, we applied a one-sided test to determine whether that difference was significant. We did the converse one-sided test for populations that appeared to have declined in fitness. Populations for which we rejected the alternate hypothesis were considered to have maintained the same fitness (i.e., 'equally fit').

## IQR variability analysis

Fitness variability was examined by plotting box-and-whisker plots of population mean fitness values, where the line, box, and whiskers represent the median, quartiles, and data within $1.5 \times$ IQR, respectively, and outlier populations beyond whiskers are shown as points (*Figure 5A*). To compare the resulting IQR for various evolution conditions and fitness assay environments, 95 % confidence intervals of the IQR were calculated from bootstrapped interval-specific replicate *s* measurements (*Figure 5B*).

To evaluate whether home environment fitness variance was less than away environment fitness variances at each evolution timepoint, we applied a Brown–Forsythe test (*Brown and Forsythe, 1974*). Since this test is typically a two-tailed test, and we wanted instead to employ a one-tailed test, we used the *z* scores from the Brown–Forsythe test to arrive at a two-tailed *t*-statistic. We could then obtain a one-tailed p value with this *t*-statistic, evaluated at $N-1$ degrees of freedom, where $N$ is the number of populations in consideration. No multiple hypothesis testing correction was applied.

## Principal components analysis

All principal components analysis excluded ancestral reference populations. To minimize the influence of varying scales of data features on the analysis, fitness values for each field – corresponding to fitness in a given assay environment, possibly at a specific evolutionary timepoint – were standardized to have a mean of 0 and standard deviation of 1 using the scikit-learn StandardScaler function. We then used the scikit-learn PCA() function.

## Clustering metric

To quantify the degree of clustering by evolution condition in the two-dimensional principal component analyses, the NearestNeighbors algorithm in the scikit-learn python package was implemented to identify the five nearest neighbors for each population in the two-dimensional PC1 versus PC2 plots (*Figure 4A and B*). The clustering metric plotted in *Figure 4D* is the number of five nearest neighbors that belong to the same evolution condition as the focal population, averaged for each evolution condition. Error bars represent 95 % confidence intervals of the mean clustering metric, which were calculated by performing the PCA and clustering analysis on bootstrapped interval-specific replicate *s* measurements. The null expectation for populations to cluster by evolution condition was computed by permuting the evolution condition 1000 times and calculating the clustering metric as described above. The true mean clustering metrics were then compared to this null expectation by calculating a multiple testing-corrected p value, computed as the percentage of permutations for which the clustering metric was greater than the true mean for a given evolution environment.

## Nonmonotonicity analysis

To assess nonmonotonicity, we linearly interpolated fitness at 500 generations for each population in each assay environment. We achieved the interpolated standard errors in fitness by taking the square root of the sum of the squares of the errors associated with the fitnesses used in the interpolation and dividing by two. For evolution plate two populations, which were offset from the others by 40 generations, we took a weighted average for the interpolation (500 generation fitness estimate) and extrapolation (1000 generation fitness estimate) steps. For the 500 generation fitness standard error estimate, we adapted this weighting approach for the standard error propagation as described for the other populations. For the 1000 generation fitness standard error estimate, we used the error assigned to the generation 960 fitness estimate. Then, we calculated the change in fitness ($\Delta s$) between 0 and 500 generations and between 500 and 1000 generations for each population in each environment, where the standard error of $\Delta s$ is propagated from the standard error of the two fitnesses used in the calculation as the square root of the sum of the squared errors. Finally, we plotted these $\Delta s$ values as $x$–$y$ coordinates. If a point and its error bars were completely within the top-left or lower-right quadrant – corresponding to an increase followed by a decrease, or a decrease followed by an increase, over the 1000-generation experiment – these were considered to be 'clearly nonmonotonic'. We applied a $\chi^2$ test with 1 degree of freedom to evaluate the significance in the difference in the frequency of nonmonotonicity in home versus away trajectories.

### Declining adaptability analysis

To assess the extent of declining adaptability among populations in their home environments, we calculated the difference in fitness for each population between 0 and 400 generations and between 600 and 1000 generations. Standard errors for these differences were calculated as the square root of the sum of the squares of the standard errors associated with the fitness estimates from each generation. Across the populations, we then compared the mean fitness change in each interval using a one-sided *t*-test, in which the alternate hypothesis was that the fitness increase in the first 400 generations was greater than the increase in the final 400 generations of the experiment.

## Acknowledgements

We thank Parris T Humphrey for assistance with experimental design and experimental protocols, and we thank Anurag Limdi for help with strain construction. We also thank Milo S Johnson for helpful comments on the manuscript.

## Additional information

### Funding

| Funder | Grant reference number | Author |
|---|---|---|
| National Defense Science and Engineering Graduate | | Christopher W Bakerlee |
| National Institutes of Health | GM007598 | Christopher W Bakerlee |
| Howard Hughes Medical Institute | Hanna H. Gray Postdoctoral Fellowship | Angela M Phillips |
| National Science Foundation | PHY-1914916 | Michael M Desai |
| National Institutes of Health | GM104239 | Michael M Desai |
| Harvard University | FAS Division of Science Research Computing Group Cannon cluster | Michael M Desai |

The funders had no role in study design, data collection and interpretation, or the decision to submit the work for publication.

### Author contributions

Christopher W Bakerlee, Angela M Phillips, Conceptualization, Formal analysis, Investigation, Methodology, Visualization, Writing – original draft, Writing – review and editing; Alex N Nguyen Ba, Conceptualization, Supervision; Michael M Desai, Conceptualization, Funding acquisition, Supervision, Writing – original draft, Writing – review and editing

### Author ORCIDs

Christopher W Bakerlee (iD) http://orcid.org/0000-0002-0819-882X
Angela M Phillips (iD) http://orcid.org/0000-0002-9806-7574
Michael M Desai (iD) http://orcid.org/0000-0002-9581-1150

### Decision letter and Author response

Decision letter https://doi.org/10.7554/eLife.70918.sa1
Author response https://doi.org/10.7554/eLife.70918.sa2

## Additional files

### Supplementary files

• Supplementary file 1. Strain creation tables.

- Supplementary file 2. Plasmid for landing pad one barcode integration.
- Supplementary file 3. Plasmid for landing pad two barcode integration.
- Supplementary file 4. Primers used in this study.
- Transparent reporting form

### Data availability

Raw amplicon sequencing reads have been deposited in the NCBI BioProject database with accession number PRJNA739738. Source data files are listed in appropriate figure legends. Analysis code is available at https://github.com/amphilli/pleiotropy-dynamics, copy archived at https://archive.softwareheritage.org/swh:1:rev:87afc41261d144c4992d7f3b4ed068b0f2c0e73d.

The following dataset was generated:

| Author(s) | Year | Dataset title | Dataset URL | Database and Identifier |
| --- | --- | --- | --- | --- |
| Bakerlee CW, Phillips AM, Nguyen Ba AN, Desai MM | 2021 | Raw sequence reads | https://www.ncbi.nlm.nih.gov/bioproject/?term=PRJNA739738 | NCBI BioProject, PRJNA739738 |
| Bakerlee CW, Phillips AM, Nguyen Ba AN, Desai MM | 2021 | Analysis code | https://github.com/amphilli/pleiotropy-dynamics | GitHub, amphilli/pleiotropy-dynamics |

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
