## [Decision Letter]

**Acceptance summary:**

When populations adapt to one environment, their fitness in that environment will increase but effects in other environments, known as pleiotropy, are unclear over longer time scales. Using a technically innovative and ambitious experimental design with evolving yeast populations, the authors show that patterns of pleiotropy depend on the evolution environment and these patterns can change over relatively short timescales. They also find a surprising amount of variation among experimental replicates that increases over time, so generalism or specialism is not deterministic.

**Decision letter after peer review:**

Thank you for submitting your article "Dynamics and variability in the pleiotropic effects of adaptation in laboratory budding yeast populations" for consideration by *eLife*. Your article has been reviewed by 3 peer reviewers, including Vaughn Cooper as Reviewing Editor and Reviewer #1, and the evaluation has been overseen by Patricia Wittkopp as the Senior Editor. The following individual involved in review of your submission has agreed to reveal their identity: G. Ozan Bozdag (Reviewer #3).

Essential revisions (for the authors):

1) All 3 reviewers found the experimental design and data intriguing, but struggled to find a clear take-home message. Most of the major suggestions in the following reviews aim to remedy this weakness. As one example, can you describe in more detail the relatively uncommon dynamics in which local adaptation corresponds to a significant decline in fitness in an away environment, such as the acetic acid populations in high- and low-temperature environments? These exceptions might prove valuable.

2) Please clarify the population sizes, because it is curious that no-stress and stressful conditions are equivalent. This might help our understanding of chance effects on within-group variation.

3) Please show how the ancestor's growth is affected by stressful conditions. If they were chosen because they generate distinct environments, what was this evidence and what is the extent of these effects?

*Reviewer #1 (Recommendations for the authors):*

The work appears to be elegant and sound, so I have nothing much to recommend in the way of more experiments. Rather I think the results (to some degree) and the discussion can help from more explicative and declarative summaries of the major findings, with an eye towards broader significance or any particular surprises you noted. This may require additional analyses of data subsets that demonstrate these novelties, but it's hard for me to pick one beyond the effects of ploidy that I mentioned.

*Reviewer #2 (Recommendations for the authors):*

While the manuscript is generally well-written and well-supported, I have one concern.

1. The authors claim that the observations in Figure 2 are consistent with the trend of declining adaptability observed in many previous studies. While this claim appears to be visually supported in the YPD+Acetic acid populations, this trend is not obvious in the other three evolution treatments (YPD, YPD 37C diploids and YPD 37C haploids). There also does not appear to be any formal statistical analysis supporting this claim. While this claim is not central to the manuscript, I would appreciate some statistical support for this statement.

*Reviewer #3 (Recommendations for the authors):*

#1 The scale of the experiment is quite impressive. However, none of the treatment groups evolved a specialist phenotype -i.e., defined with the evolution of a pleiotropic outcome where replicate populations as a whole start to lose against their ancestor in an alternate environment. There is one case where adaptation to acetic acid corresponds to a decline in median fitness in low-temperature, but this is not true for all populations as some gain fitness in the away environment. It would have been fascinating to observe the dynamics of specialization in an experimental setup like this (e.g., is there a monotonic or a fluctuating decline in fitness in the away environment?). Considering the large scale of these experiments, I will not advise testing the temporal dynamics of pleiotropy across other away environments (e.g., low or high pH, galactose) with the hope of discovering an away environment where there is a corresponding cost of adaptation. However, I still would prefer authors to look at some instances where local adaptation corresponds to a significant decline in fitness in an away environment. A large number of the acetic acid populations show a fitness decline in high- and low-temperature environments. Extracting this data and discussing the trends in the evolution of fitness costs would be pretty valuable.

#2 A numerical summary of the pleiotropic outcomes is worth considering. For instance, how many populations showed higher relative fitness gains in an away environment, how many populations showed a decline in relative fitness in an away environment, and how many populations showed similar relative fitness outcomes between the home and an away environment. Doing this comparatively for generation-200 and generation-1000 would be interesting. The number of replicate populations is a strength of this study, and such a numerical summary would be helpful for a potential future review paper summarizing the results of pleiotropy papers.

#3 Jerison et al., (2020) lately tested the pleiotropic consequences of local adaptation using the same model organism for 700 experimental evolution. They report that specialization is common in high salt (!) and low and high pH environments. However, here we see a contradictory result for the high salt environment. Instead of a decrease, there is an increase in fitness across all evolution treatments. It would have a general value to discuss this contrasting outcome in the sign of pleiotropic effects despite the growth of the same model organism under quite similar growth conditions. The authors say that they chose these environments to facilitate comparisons with previous studies in yeast, but they do not come back to this point.

#4 The pleiotropy is tested in three (plus two) different environments. The difference among environments is a crucial part of an article that focuses on pleiotropy. These environments are said to be different as they are supposed to apply distinct types of stress. Therefore, it is necessary to quantify the negative effect of stress on an ancestral population, ideally with a small-scale relative fitness assay or a growth curve measurement in the same well-plate setup (e.g., YPD+0.2% acetic acid vs. YPD).

The authors cite two papers regarding the choice of these stressful environments. However, I could not find a quantitative assay reporting the effect of stress in those papers (e.g., see Methods section in Jerison et al., 2020).

#5 The study reports a large variability in pleiotropic effects. It would be beneficial to report the population size and bottleneck size for each condition, at least for the early stages of this experiment. The demographic differences would affect the interplay between drift and selection differently across the evolution treatments, potentially providing more insights about the results.

Following this, the authors report a daily bottleneck of 10,000 cells for each environment (each culture is transferred by the pipetting robot). It means that there is no difference in population size at saturation across environments. If this is true, the stress conditions do not negatively affect the ancestral population. Alternatively, the population size estimates do not represent the reality of each environmental condition (e.g., high temperature is expected to impact the growth). Therefore, I recommend carefully measuring the bottleneck size in five conditions for a few founder clones, and if Ne values are significantly different, please report the outcome in the main text.

#6 The study presents a large dataset and numerous plots, with varying outcomes for most experiments (# of plots: 20 in Figure 2, 10 in Figure 3, 16 in Figure 4, 24 in Figure 5, 8 in Figure 6). However, it is laborious to connect all those different visual information with a take-home message in the end. Of course, the Discussion section summarizes the results nicely, but the paper would still benefit from a graphical, conceptual summary. For instance, using the plots in Figure 3 as a template (i.e., comparing fitness trajectories in-home vs. away environment) and/or Figure 4A, it would be nice to summarize the most important findings of this work. A conceptual summary that focuses on the characteristic and uncharacteristic findings of the study and how that changes the way we understand pleiotropy.

#7 Figure 2 shows the temporal dynamics of fitness change. Even though each plot has a bold black line showing the median fitness, it is hard for the reader to compare relative fitness at time-0 and time-1000 quantitatively. Despite that large variability across populations, it would still be valuable to report the median relative fitness at the final time point in the main text. Looking at Figure 2, it is not possible to tell whether populations evolving in high temperatures reach a higher fitness than their ancestor. For instance, see Figure 1 in Jerison et al., 2020 for a nice graphical summary.

---

## [Author Response]

Essential revisions:1) All 3 reviewers found the experimental design and data intriguing, but struggled to find a clear take-home message. Most of the major suggestions in the following reviews aim to remedy this weakness. As one example, can you describe in more detail the relatively uncommon dynamics in which local adaptation corresponds to a significant decline in fitness in an away environment, such as the acetic acid populations in high- and low-temperature environments? These exceptions might prove valuable.

We agree with the overall comment and have addressed all of the reviewers’ major suggestions on this point (e.g. by summarizing the changes in fitness in home and away environments; Figure 2B and Figure 2—figure supplements 2–5). We have also expanded our discussion of general trends as well as uncommon dynamics and the variability of outcomes.

2) Please clarify the population sizes, because it is curious that no-stress and stressful conditions are equivalent. This might help our understanding of chance effects on within-group variation.

We have added the population bottleneck sizes in each evolution and assay environment to Figure 1–source data 1. These bottleneck sizes are largely consistent across evolution and assay environments, because populations ultimately reach similar saturation densities between the daily (or in the case of the 21°C environment, 48-hour) 1:1024 dilutions. However, we can see clear differences among the growth curves in the different environments (Figure 1—figure supplement 2).

3) Please show how the ancestor's growth is affected by stressful conditions. If they were chosen because they generate distinct environments, what was this evidence and what is the extent of these effects?

We have now measured the growth of the ancestral strains (haploid and diploid) in each of the evolution and assay environments, and provide these data in Figure 1—figure supplement 2. Diploids and haploids reach saturation most quickly at 37°C and in YPD, followed by NaCl and acetic acid, and finally at 21°C. These environments were chosen because they cause distinct types of physiological stress, as evidenced by previous evolution experiments and fitness measurements (e.g. Nguyen Ba et al., 2019, Jerison et al., 2020, Kinsler et al., 2020) and by phenotypic studies in yeast (e.g., Giannattsio et al., 2013, Taymaz-Nikerel et al., 2016) and other organisms (e.g., Trček et al., 2015, Lamitina et al., 2004, Fasolo and Krebs, 2004). In addition, the distinct fitness trajectories (Figure 2) and trade-offs (Figure 3) in this work provide retrospective evidence for the differences between the biological effects of these environments.

Reviewer #1 (Recommendations for the authors):The work appears to be elegant and sound, so I have nothing much to recommend in the way of more experiments. Rather I think the results (to some degree) and the discussion can help from more explicative and declarative summaries of the major findings, with an eye towards broader significance or any particular surprises you noted. This may require additional analyses of data subsets that demonstrate these novelties, but it's hard for me to pick one beyond the effects of ploidy that I mentioned.

As described above, to more clearly show general trends and variation in pleiotropy, we have added figure panels summarizing the changes in fitness across all populations (Figure 2B and Figure 2—figure supplements 2–5). We have also expanded our discussion of these trends and their variability, and our description of particular examples as suggested here and in other reviewer comments.

Reviewer #2 (Recommendations for the authors):While the manuscript is generally well-written and well-supported, I have one concern.1. The authors claim that the observations in Figure 2 are consistent with the trend of declining adaptability observed in many previous studies. While this claim appears to be visually supported in the YPD+Acetic acid populations, this trend is not obvious in the other three evolution treatments (YPD, YPD 37C diploids and YPD 37C haploids). There also does not appear to be any formal statistical analysis supporting this claim. While this claim is not central to the manuscript, I would appreciate some statistical support for this statement.

Thanks, this is a good point. We now provide statistical support for this statement (for those cases where it is in fact significant) in Figure 2-figure supplement 6.

Reviewer #3 (Recommendations for the authors):#1 The scale of the experiment is quite impressive. However, none of the treatment groups evolved a specialist phenotype -i.e., defined with the evolution of a pleiotropic outcome where replicate populations as a whole start to lose against their ancestor in an alternate environment. There is one case where adaptation to acetic acid corresponds to a decline in median fitness in low-temperature, but this is not true for all populations as some gain fitness in the away environment. It would have been fascinating to observe the dynamics of specialization in an experimental setup like this (e.g., is there a monotonic or a fluctuating decline in fitness in the away environment?). Considering the large scale of these experiments, I will not advise testing the temporal dynamics of pleiotropy across other away environments (e.g., low or high pH, galactose) with the hope of discovering an away environment where there is a corresponding cost of adaptation. However, I still would prefer authors to look at some instances where local adaptation corresponds to a significant decline in fitness in an away environment. A large number of the acetic acid populations show a fitness decline in high- and low-temperature environments. Extracting this data and discussing the trends in the evolution of fitness costs would be pretty valuable.

We agree that these cases of specialization and evolution of fitness costs are particularly interesting. As described above, to more clearly show general trends and variation in pleiotropy, including particularly the cases where fitness declines do occur, we have summarized the changes in fitness across all populations in Figure 2B and Figure 2—figure supplements 2–5. We have also added a paragraph to the Results discussing specialization.

#2 A numerical summary of the pleiotropic outcomes is worth considering. For instance, how many populations showed higher relative fitness gains in an away environment, how many populations showed a decline in relative fitness in an away environment, and how many populations showed similar relative fitness outcomes between the home and an away environment. Doing this comparatively for generation-200 and generation-1000 would be interesting. The number of replicate populations is a strength of this study, and such a numerical summary would be helpful for a potential future review paper summarizing the results of pleiotropy papers.

As described above, we have performed this analysis for each 200-generation interval and present the corresponding data in Figure 2B and Figure 2—figure supplements 2–5.

#3 Jerison et al., (2020) lately tested the pleiotropic consequences of local adaptation using the same model organism for 700 experimental evolution. They report that specialization is common in high salt (!) and low and high pH environments. However, here we see a contradictory result for the high salt environment. Instead of a decrease, there is an increase in fitness across all evolution treatments. It would have a general value to discuss this contrasting outcome in the sign of pleiotropic effects despite the growth of the same model organism under quite similar growth conditions. The authors say that they chose these environments to facilitate comparisons with previous studies in yeast, but they do not come back to this point.

We agree that the difference between these observations is noteworthy, and we now compare these results in the discussion. In particular, we note that there are several differences between these ancestral strains we use here compared to the Jerison study (e.g. Jerison et al., 2020 uses a W303-derived strain, which is haploid, differs at key fitness-determining loci such as MKT1, and contains killer virus). In addition, there are also some differences in the evolution experiments (e.g. different concentrations of salt). The apparent dependence of the results of these types of experiments on strain background (and other factors) further highlights the extensive variability in the pleiotropic outcomes of adaptation.

#4 The pleiotropy is tested in three (plus two) different environments. The difference among environments is a crucial part of an article that focuses on pleiotropy. These environments are said to be different as they are supposed to apply distinct types of stress. Therefore, it is necessary to quantify the negative effect of stress on an ancestral population, ideally with a small-scale relative fitness assay or a growth curve measurement in the same well-plate setup (e.g., YPD+0.2% acetic acid vs. YPD).

The authors cite two papers regarding the choice of these stressful environments. However, I could not find a quantitative assay reporting the effect of stress in those papers (e.g., see Methods section in Jerison et al., 2020).

As noted above, we now provide growth curves for ancestral haploid and diploid clones in each of the evolution and assay environments (Figure 1—figure supplement 2). In addition, we have added additional references discussing the effects of these and other stresses on budding yeast.

The authors cite two papers regarding the choice of these stressful environments. However, I could not find a quantitative assay reporting the effect of stress in those papers (e.g., see Methods section in Jerison et al., 2020).#5 The study reports a large variability in pleiotropic effects. It would be beneficial to report the population size and bottleneck size for each condition, at least for the early stages of this experiment. The demographic differences would affect the interplay between drift and selection differently across the evolution treatments, potentially providing more insights about the results.Following this, the authors report a daily bottleneck of 10,000 cells for each environment (each culture is transferred by the pipetting robot). It means that there is no difference in population size at saturation across environments. If this is true, the stress conditions do not negatively affect the ancestral population. Alternatively, the population size estimates do not represent the reality of each environmental condition (e.g., high temperature is expected to impact the growth). Therefore, I recommend carefully measuring the bottleneck size in five conditions for a few founder clones, and if Ne values are significantly different, please report the outcome in the main text.

As noted above, we have provided the population bottleneck sizes for ancestral clones in each of the evolution and assay environments in Figure 2—figure supplement 2. These bottleneck sizes are quite similar across environments and thus the interplay between drift and selection should not vary between evolution environments.

Although the population size does not vary substantially between evolution and assay environments, there are substantial differences in the corresponding patterns of population growth, reflecting the differentially stressful nature of these conditions. These growth curves are now provided in Figure 2—figure supplement 3.

#6 The study presents a large dataset and numerous plots, with varying outcomes for most experiments (# of plots: 20 in Figure 2, 10 in Figure 3, 16 in Figure 4, 24 in Figure 5, 8 in Figure 6). However, it is laborious to connect all those different visual information with a take-home message in the end. Of course, the Discussion section summarizes the results nicely, but the paper would still benefit from a graphical, conceptual summary. For instance, using the plots in Figure 3 as a template (i.e., comparing fitness trajectories in-home vs. away environment) and/or Figure 4A, it would be nice to summarize the most important findings of this work. A conceptual summary that focuses on the characteristic and uncharacteristic findings of the study and how that changes the way we understand pleiotropy.

As noted above, we have summarized changes in fitness across environments in Figure 2B and Figure 2—figure supplements 2-5, and have expanded our discussion of the trends and exceptions in these data.

#7 Figure 2 shows the temporal dynamics of fitness change. Even though each plot has a bold black line showing the median fitness, it is hard for the reader to compare relative fitness at time-0 and time-1000 quantitatively. Despite that large variability across populations, it would still be valuable to report the median relative fitness at the final time point in the main text. Looking at Figure 2, it is not possible to tell whether populations evolving in high temperatures reach a higher fitness than their ancestor. For instance, see Figure 1 in Jerison et al., 2020 for a nice graphical summary.

We have addressed this change in a few different ways to make Figure 2 easier to digest. First, we have added a y = 0 line; second, we note the median fitness at generation 1000 in each plot; third, we have colored the lines so that the median line is more legible. We also provide a summary of fitness changes in Figure 2B and Figure 2—figure supplements 2-5.